# Anomalous entropy-driven kinetics of dislocation nucleation

Soumendu Bagchi [1,2] & Danny Perez [1]

The kinetics of dislocation reactions, such as dislocation multiplication, controls the plastic deformation in crystals beyond their elastic limit, therefore critical mechanisms in a number of applications in materials science. We present a series of large-scale molecular dynamics simulations that shows that one such type of reactions, the nucleation of dislocation at free surfaces, exhibit unconventional kinetics, including unexpectedly large nucleation rates under compression, very strong entropic stabilization under tension, as well as strong non-Arrhenius behavior. These unusual kinetics are quantitatively rationalized using a variational transition state theory approach coupled with an efficient numerical scheme for the estimation of vibrational entropy changes. These results highlight the need for a variational treatment of the kinetics to quantitatively capture dislocation reaction kinetics, especially at low-to-moderate strains where large deformations are required to activate reactions. These observations suggest possible explanations to previously observed unconventional deformation kinetics in both molecular dynamics simulations and experiments.

Although dislocation multiplication is one of the key mechanisms controlling the plastic deformation of materials, a complete understanding of the exact nature of the multiplication mechanism and their corresponding kinetics remains elusive[1]. Preexisting glissile defects (e.g., Frank-Read sources[2]) are usually assumed to be the most efficient dislocation sources in typical metal microstructures. However, nucleation of new dislocations at heterogeneities, e.g, ledges or steps in free surfaces[3,4], crack-tips in single crystals[5,6] and grain/twin boundary ledges in polycrystals[7] can also lead to dislocation multiplication, leading to debates on the relative importance of the different mechanisms[4]. In contrast, the situation in defect-starved pristine crystals such as micro/nano-pillars is clearer, as it is generally accepted that nucleation of new dislocations at free surfaces will dominate[8], given the lack of suitable internal nucleation sites.

While the very high energy barriers predicted for heterogeneous nucleation-driven multiplication at commonly accessible stresses[5,9,10] could be taken as an indication that this process is unlikely to be competitive when other multiplication sources are available, recent insights suggest that energetic considerations alone might only paint a partial picture of the kinetics of dislocation reactions. For example, strong entropic effects have been computationally observed in a range of processes, including homogeneous nucleation of dislocation loops[11], transformation from vacancy clusters to stacking-fault tetrahedra[12], growth of nano-voids under tensile stress[13], dislocation emission from crack-tips[6], and during nano-indentation[14]. While the importance of entropic effects is increasingly clear, these typically cannot be reliably characterized using conventional static methods such as harmonic transition state theory[15], and instead require free-energy-based treatments[16,17] that can be extremely computationally demanding, especially for the large system sizes typical of dislocation studies. This limits systematic investigations of key reaction processes and encourages the use of heuristics to extrapolate to regimes that are inaccessible to direct simulation.

To address this challenge, we leverage a simple and computationally efficient analytic approximation to the activation entropy of the dislocation reaction that enables a variational treatment of the reaction kinetics. This approach predicts that the large static energy barriers for surface nucleation of dislocations can be offset by

[1]Theoretical Division, Los Alamos National Laboratory, Los Alamos, NM, USA. [2]Center for Nanophase Materials Sciences, Oak Ridge National Laboratory, Oak Ridge, TN, USA. ✉e-mail: bagchis@ornl.gov

considerable vibrational-entropic contributions in systems loaded in compression. The rapid increase in vibrational entropy as dislocation loops nucleate and grow (caused by the softening of the phonon modes perpendicular to the reaction coordinate that follows from the partial release of compressive strains) leads to a dramatic reduction in the activation free-energy barrier, in turn causing a colossal increase in predicted nucleation rate compared to conventional "standard pre-factor" assumptions. In contrast, systems loaded in tension show the opposite effect: the phonon stiffening that occurs during the release of tensile strain instead leads to an entropic suppression of nucleation. In both cases, the temperature-dependent location of the kinetic bot-tleneck for nucleation leads to strongly non-Arrhenius effects, high-lighting the need for variational treatments of the kinetics. This approach opens the door to routine investigations of anharmonic entropic effects in dislocation reactions using atomistic simulations, which could have far-reaching consequences on multiscale modeling of thermally-activated events like kink-pair nucleation/migration[18,19], slip nucleation/transmission from/across grain boundaries[20], cross-slip[21], twin nucleation/migration[22] events as well as formation and destruction of dislocation junctions, all of which play important roles in the plasticity of metals.

In the following, we consider the kinetics of dislocation nucleation events from surface ledges of fcc copper. Direct MD simulations of nucleation and static harmonic transition state theory kinetics are first shown to be incompatible. These discrepancies are then reconciled

using a variational treatment that accounts for the changing location of the kinetic bottleneck with temperature, using an analytic approx-imation to the entropy variation along the static pathway. The fol-lowing sections address the asymmetry in compressive and tensile loading, which leads to exceptionally high and low effective prefactors, respectively, especially at high temperature and low strains, and show that the variational rate theory captures the non-Arrhenius tempera-ture behavior. We finally relate our main finding to previous compu-tational and experimental observations, before concluding.

## Results

### Thermal activation of dislocation nucleation under compression

In the first set of simulations, we consider the nucleation of dis-locations under uniaxial compressive strains. Figure 1 shows the gradual nucleation and growth of a dislocation half-loop at a pre-existing surface step that was introduced to act as a stress con-centrator for an applied strain of 2%. An ensemble of 50 large-scale simulations containing 38 million atoms was subjected to a slow temperature increase at a rate of 10 K/ns between T = 600 K and 700 K (c.f., "Methods" section). In each case, a partial dislocation nucleated from the preexisting surface step at times that ranged from 0.5 ns to 5.0 ns (c.f. Supplementary Fig. 1). Configurations extracted from one of the nucleation events were then used to seed a calculation of the static minimum energy pathway (MEP) associated with the nucleation process using a multi-step climbing-image

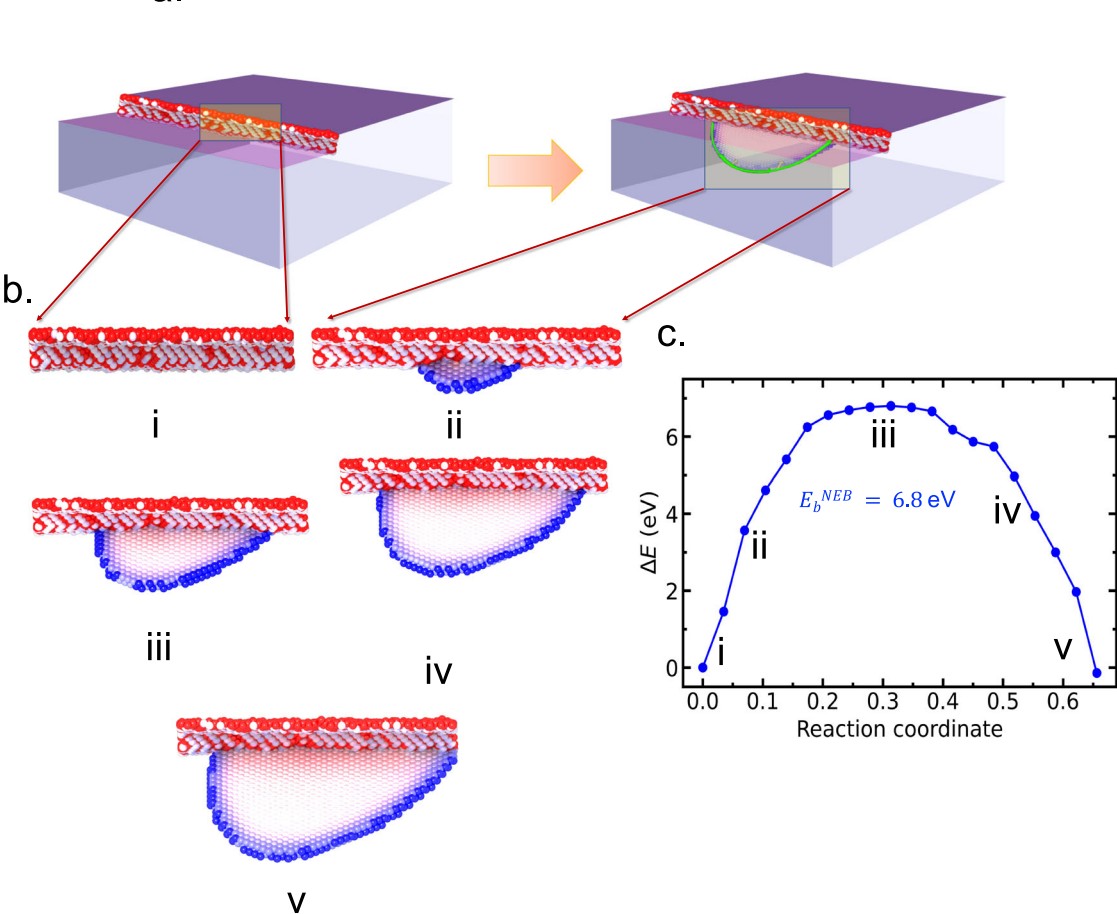

**Fig. 1 | Nucleation of a dislocation half-loop under an applied compressive strain. a** Atomistic model of fcc Cu single crystal where a surface step acts as a stress concentrator and dislocation nucleation site. **b** Nucleation and expansion of a leading partial dislocation half-loop into the bulk under 2% uniaxial compression. Configurations (i–v) are samples along the minimum energy path (**c**) associated

with the process. The atoms are colored based on their local centrosymmetry parameter. Core atoms in the half loop partial are colored blue, which enclose a stacking fault. Configuration (iii) corresponds to the saddle point along the mini-mum energy pathway.

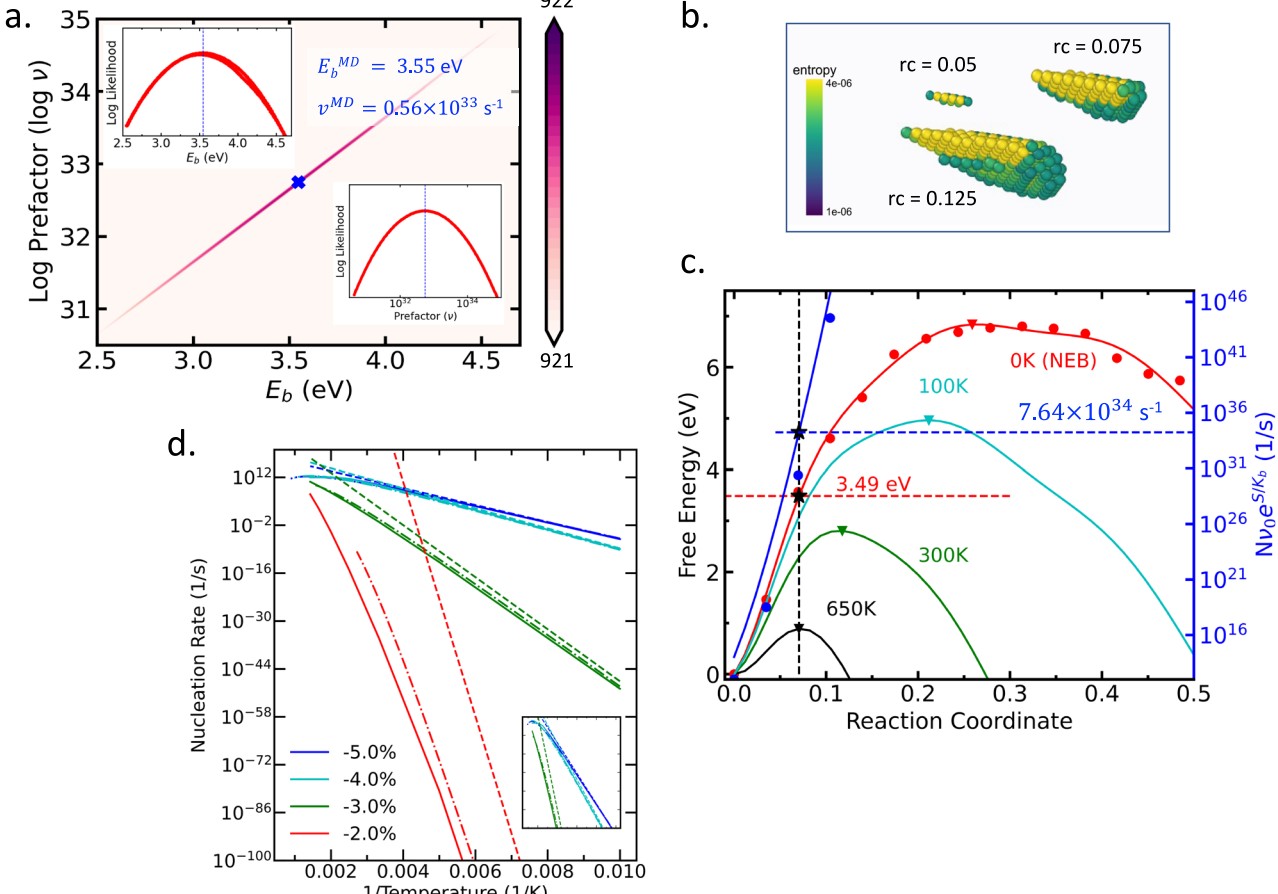

**Fig. 2 | Nucleation kinetics under 2% compression. a** Contour of Log-likelihood of nucleation events as a function of two canonical TST-based kinetic model parameters (i) activation energy ($E_b$) and (ii) frequency prefactor ($\nu$) computed from a series of large scale atomistic simulations. The maximum likelihood region is concentrated on a narrow band in the parameter-space with optimal (blue dashed lines inset) activation energy ($E_b$ shown inset) and frequency prefactor ($\nu$ inset) as 3.55 eV and $5.57 \times 10^{32}$ s$^{-1}$. **b** Per-atom entropy contribution at different locations along the reaction coordinates show that entropy increases originate from atoms in the faulted region. **c** The free-energy profiles along the MEP reveal a temperature-dependent maxima location that shifts towards smaller loops and lowers free-energy barriers as temperature increases. The black star pointers indicate the nucleation prefactor (on the blue curve) and nucleation static PE barrier (red curve) on the location of the maximum of the free energy curve (black) at nucleation temperature (650K). **d** Nucleation rate predictions under different levels of compressive strain. Solid lines represent the full variational TST predictions, while dashed-dotted and dashed lines show predictions from an analytical approximation to variational TST (c.f. Sec. 1.5) and HTST estimations, respectively. When not distinguishable, the full and approximate variational TST treatments overlap with one another.

nudged elastic band (CI-NEB) computational protocol (c.f. "Method" section).

As the half-loop grows (Fig. 1c), the energy of the system steadily increases until a critical radius (~33Å) is reached. At this point, the cost of growing the loop further is offset by an equal decrease in the elastic strain energy stored in the system (b, iii). Past this point, the loop becomes super-critical and spontaneously grows (b iv, v) until it is absorbed at the other free surface. At 2% compression, the corresponding energy barrier for nucleation is very high at 6.8 eV, consistent with previous estimates from linear elasticity and atomistic studies[3,5,23,24]. Assuming the kinetics to be locally Arrhenius (i.e., $k = \nu \exp(-E_b/kT)$) over the temperature range from 600 to 700 K and that the relevant energy barrier is the one measured in the MEP ($E_b^{\text{NEB}} = 6.8$ eV), the average value of the nucleation times observed in MD simulations suggests an extraordinarily high prefactor of $\nu \simeq 10^{61}$/s, which sharply contrasts with "standard prefactor" values of around $10^{12}$/s multiplied the number of equivalent nucleation sites along the step (120 in this case).

In contrast, a joint likelihood analysis for effective Arrhenius parameters $E_b^{MD}$ and $\nu^{MD}$ conditioned on the observed distribution of nucleation times measured in MD simulations (c.f., Fig. 2a and Sec.??) instead suggests maximum likelihood estimates of $E_b^{MD} \simeq 3.55$ and

$\nu^{MD} \simeq 0.56 \times 10^{33}$/s. As shown Fig. 2a, the posterior distribution unambiguously excludes the static barrier value $E_b^{\text{NEB}} = 6.8$ eV. These results show a clear tension between kinetics inferred from MD simulations and a conventional treatment.

## Variational transition state theory

As the MEP was obtained directly from reactions observed in MD simulations, the dramatic disagreement between $E_b^{\text{NEB}}$ and $E_b^{\text{MD}}$ suggests that significant changes in entropy must occur along the transition pathway in such a way that the kinetic bottleneck for dislocation nucleation does not coincide with the crossing of the hyperplane perpendicular to the energy saddle, as commonly assumed in conventional harmonic transition state theory (HTST)[15].

While methods have been proposed to directly locate minimum free-energy pathways[16,17,25], applications to large systems remain challenging and require extensive simulations at each temperature, which would be extremely challenging for the system sizes of interest here. We instead follow a simplified approach where we assume that the MEP provides a good reaction coordinate for reactions at finite temperature, but that the location of the optimal dividing surface along the MEP is temperature dependent. According to the variational

formulation of TST[26] (which follows from the fact that TST provides an upper bound to the true transition rate), the optimal dividing surface is the one that minimizes the TST escape rate. In our context, this surface is approximated by a hyperplane perpendicular to the MEP passing through the (temperature-dependent) free-energy maximum at the temperature of interest. The conventional HTST treatment corresponds to the low-temperature limit where the optimal hyperplane passes through the energy saddle point. In this formalism, the nucleation rate becomes

$$k(T) \simeq N_{sites} \nu_0 \exp \frac{-(F(\xi^*(T)) - F(\xi = 0))}{k_B T}$$
$$= N_{sites} \nu_0 \exp\left[S(\xi^*(T)) - S(\xi = 0)\right] \quad (1)$$
$$\exp \frac{-(V(\xi^*(T)) - V(\xi = 0))}{k_B T}$$

where $\xi^*(T)$ corresponds to the location of the free-energy maximum at temperature $T$ and $\nu_0$ is a so-called standard prefactor corresponding to the frequency of a typical phonon mode, and the multiplicity factor $N_{sites}$ accounts for degenerate nucleation pathways that are equivalent by symmetry. This simplified variational treatment becomes tractable providing an estimate of the change in vibrational entropy $\Delta S(\xi) = S(\xi) - S(0)$ along the MEP. Note that, strictly speaking, the free energy in the above expression is obtained by integrating over only degrees of freedom *perpendicular* to the reaction coordinate.

## Entropy and free energy estimation

A computationally efficient approximation of the entropy change can be obtained in a continuum thermodynamic setting. Consider the change in free energy density due to a small deformation field as $\Delta F = -\frac{1}{2}\sigma_{ij}\epsilon_{ij}$, where $\sigma_{ij}$ and $\epsilon_{ij}$ are the stress and Green-Lagrange strain tensors, respectively.

Considering linear elasticity ($\sigma_{ij} = C_{ijkl}\epsilon_{kl}$) at finite temperature and linear thermal expansion ($\alpha_{ij} = \frac{\partial \epsilon_{ij}}{\partial T}$), the volumetric entropy density can be derived from the partial variation of the free energy with respect to temperature:

$$s = \frac{\partial \Delta F}{\partial T} = -\frac{1}{2}\left(\frac{\partial \sigma_{ij}}{\partial T}\epsilon_{ij} + \sigma_{ij}\frac{\partial \epsilon_{ij}}{\partial T}\right)$$
$$= -\frac{1}{2}\left(\frac{\partial (C_{ijkl}\epsilon_{kl})}{\partial T}\epsilon_{ij} + \sigma_{ij}\alpha_{ij}\right)$$
$$= -\frac{1}{2}\left(\frac{\partial C_{ijkl}}{\partial T}\epsilon_{ij}\epsilon_{kl} + C_{ijkl}\frac{\partial \epsilon_{kl}}{\partial T}\epsilon_{ij} + \sigma_{ij}\alpha_{ij}\right) \quad (2)$$
$$= -\frac{1}{2}\left(\frac{\partial C_{ijkl}}{\partial T}\epsilon_{ij}\epsilon_{kl} + 2\sigma_{ij}\alpha_{ij}\right)$$

For a material with an isotropic thermal expansion $\alpha_{ij} = \alpha\delta_{ij}$ and linear bulk volumetric deformation ($\sigma_h = -\sigma_{ii}/3 = K\epsilon_{ii}$), these expressions simplify to:

$$s = -\frac{1}{2}\left(\frac{\partial C_{ijkl}}{\partial T}\epsilon_{ij}\epsilon_{kl} + 2\alpha\sigma_{ii}\right)$$
$$= -\frac{1}{2}\frac{\partial C_{ijkl}}{\partial T}\epsilon_{ij}\epsilon_{kl} + 3\alpha\sigma_{ii}/3 \quad (3)$$
$$= -\frac{1}{2}\frac{\partial C_{ijkl}}{\partial T}\epsilon_{ij}\epsilon_{kl} + 3\alpha K\epsilon_{ii}$$

From this last expression, the entropy difference between two configurations that differ by an elastic distortion indexed by a variable $\xi$ along a reaction pathway can be obtained through a volume integration of the entropy density:

$$\Delta S(\xi) = 3\alpha K \int_{V(\xi)} \epsilon_{ii}(\xi)dV - \frac{1}{2}\frac{\partial C_{ijkl}}{\partial T}\int_{V(\xi)} \epsilon_{ij}(\xi)\epsilon_{kl}(\xi)dV \quad (4)$$

For a rigorous derivation based on higher-order finite-strain elasticity, c.f.[27].

The first term captures the effect of variations in the net volume occupied by the solid, while the second term captures contributions of higher-order elastic distortions. Fundamentally, this entropy change stem from the anharmonicity of the potential energy which lead to a strain dependence of the phonon frequencies. This in turns leads to thermal expansion ($\alpha$) and elastic softening $\left(\frac{dC_{ijkl}}{dT}\right)$.

In order to map the continuum formulation into atomistic configuration, the integration over volume is expressed as a sum over per-atom quantities multiplied by their respective Vononoi volume. Unbounded volumes corresponding to surface atoms are excluded from the sum. The per-atom Green-Lagrange strain is estimated following Ref.? using a least-square minimization of the residual between actual atomistic displacements and an affine local displacement measure defined in a continuum setting with a neighbor cutoff of $10\text{Å}$ per atom. All the material parameters (e.g. elastic coefficients $C_{ijkl}$, $K$, $\alpha$, and their temperature derivatives) correspond to Mishin's[28] EAM force field that was used in the simulations.

Note that this entropy formulation does not resolve the contribution of degrees of freedom that are parallel and perpendicular to the MEP. When using this formula in conjunction with Eq. (1), one therefore implicitly assume that the parallel contributions cancel exactly along the MEP. This is expected to be a very accurate approximation, as the contributions from individual modes to the total entropy changes are typically very small; instead, it is the summation of a very large number of small contributions from each perpendicular mode that leads to significant entropy change along the reaction coordinate[13].

The variation in entropy and the corresponding free energy profiles relative to the original energy minimum ($\xi = 0$), are reported in Fig. 2b). As the half-loop grows, $\Delta S$ sharply increases due to the gradual relaxation of elastic compressive strains around the faulted region, leading to local softening and consequently to a steady increase in vibrational entropy as the system progresses along the MEP. The inset of Fig 2a confirms that the leading (positive) entropic contributions indeed localize close to the leading partial dislocation half-loop. This increase in vibrational entropy along the MEP dramatically affects the free-energy landscape as the temperature increases, gradually decreasing the free-energy barrier and shifting the free-energy maximum towards smaller critical loops.

## Nucleation kinetics

The variational treatment leads to two key observations: (i) the HTST prefactors, corresponding to the low-temperature limit, are extremely large compared to standard values, and (ii) the displacement of the kinetic bottleneck away from the energy saddle plane leads to a non-Arrhenius temperature dependence (c.f., Fig. 2c). Indeed, the HTST prefactors are estimated to be $\sim10^{124}$/s, $\sim10^{30}$/s, and $\sim10^{22}$/s for 2,3, and 4% compression, respectively, which is tens to hundreds of orders of magnitude higher than standard values. This clearly shows that an interpretation of the HTST prefactor in terms of an attempt frequency is completely inappropriate here, as the value of the prefactor instead results from the collective contribution from a large number of vibrational modes whose frequency individually varies by a small amount.

The results also show that the variational rate theory leads to non-Arrhenius corrections that decrease the rate compared to the HTST baseline, which is expected since the free-energy barrier can only variationally increase. The variational prediction from Eq. (1) also

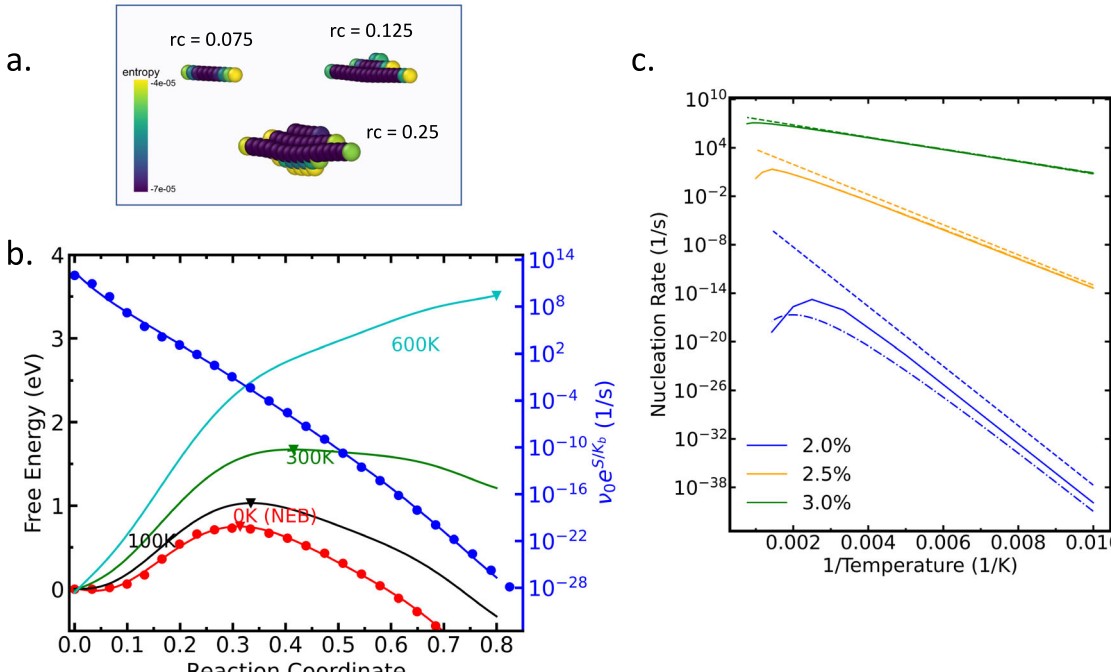

**Fig. 3 | Entropic suppression of nucleation under tension. a** Per-atom entropy contributions under 2% tension. **b** Evolution of the free-energy profiles with temperature. Contrary to the case of compression, vibrational entropy decreases along the MEP, leading to the stabilization of the pristine state. The inverted triangles indicate maxima (variational saddle) of the free energy at various temperatures. **c** Estimated nucleation rates ($\nu_0 e^{-\Delta F/k_B T}$) inferred from the variational free energy

barriers. Solid lines represent the full variational TST predictions, while dashed-dotted and dashed lines show predictions from an analytical approximation to variational TST (c.f. Sec. 1.5) and HTST estimations, respectively. When not distinguishable, the full and approximate variational TST treatments overlap with one another.

locally agrees with the effective Arrhenius model inferred from the MD data (c.f., Fig. 2b). The predicted energetic component of the barrier at 650K is indeed 3.49 eV (compared to 3.55 eV for the effective model) and the predicted prefactor $7.64 \times 10^{34}$/s (compared to $0.56 \times 10^{33}$/s for the effective model), demonstrating that the simple entropy estimation method proposed above captures the essential physics of the process observed in MD. The non-Arrhenius effects become less pronounced at higher strains (3% or more), as the steady decrease of the critical half-loop size leads to smaller changes in the elastic distortion between the minimum and energy saddle point and consequently to a lesser vibrational-entropic contribution to the kinetics. In this regime, the HTST predictions become adequate over most of the temperature range, although deviations are still noticeable at higher temperatures and moderate strains.

### Thermal activation of dislocation nucleation under tension

A dramatically different behavior is observed in systems loaded under uniaxial tension (c.f., Fig. 3a). In this case, the growth of the half-loop releases tensile strain, leading to a gradual stiffening of the phonon modes perpendicular to the MEP, and hence to a vibrational entropy decrease as the loop nucleation proceeds. This translates into a significant increase in the free energy barrier for nucleation with increasing temperature and to a displacement of the free energy maximum along the MEP towards larger loops (c.f., Fig. 3a for a 2% tensile loading). Thus, in contrast to the case of compressive loading where vibrational entropy promotes nucleation, vibrational entropy stabilizes the pristine state and inhibits nucleation under tension, leading to extremely low HTST prefactors that range between $10^{-5}$ and $10^{-2}$/s (Fig. 3a), about ten orders of magnitude lower than standard values. Once again, non-Arrhenius effects are most pronounced at low strains, where the critical loops are large, and at high temperatures, where the variational displacement of the kinetic bottleneck becomes significant.

When entropic contributions are sufficiently large, the (variational) value of $\Delta F(T)/k_B T$ can even increase with temperature, leading to a predicted anti-Arrhenius temperature dependence of the nucleation rate. These predictions are confirmed by direct MD simulations under a uniaxial strain rate of $2 \times 10^8$/s. Indeed, in near-athermal conditions i.e. ~10 K, a dislocation is nucleated (as indicated by a drop in potential energy curve shown in Fig. 4a) at a strain $\epsilon_{atherm} = 3.8\%$; this strain corresponds to the conventional mechanical instability limit where the energy barrier for nucleation vanishes. At relatively low temperatures (100K–200K), nucleation is observed at strains lower than $\epsilon_{atherm}$ (i.e. 3% and 2.8%), in qualitative agreement with conventional thermally-activated kinetics. However, as the temperature further increases beyond 300K, the trend reverses and nucleation is observed at increasingly large strains that exceed the athermal limit (e.g., $\epsilon_{300K} = 4.75\%$) This trend continues (Fig. 4a) for even higher temperature (500K), where nucleation is suppressed up to 5% tensile strain, indicating a very strong entropic stabilization. This observation is consistent with the free-energy analysis of the nucleation process. Indeed, at strains exceeding the athermal limit, the MEP for nucleation of a half loop is barrier-less and steeply downhill in energy (c.f. T = 0 K line in Fig. 4b for a tensile strain of 5%). At higher temperatures (T > 350 K) however, the pristine initial state becomes entropically stabilized by extremely large free-energy barriers that opposes nucleation (c.f. T = 500 K in Fig. 4c). These results are consistent with previous reports of anti-Arrhenius behavior in MD simulations of dislocation emission from nanoscale voids[13] and of entropic suppression of homogeneous dislocation nucleation in nanoindentation simulations[14].

## Discussion

Our results demonstrate that extremely strong entropic contributions can be expected in a broad range of situations where reaction coordinates strongly couple to long-range strain fields, e.g., when the strain

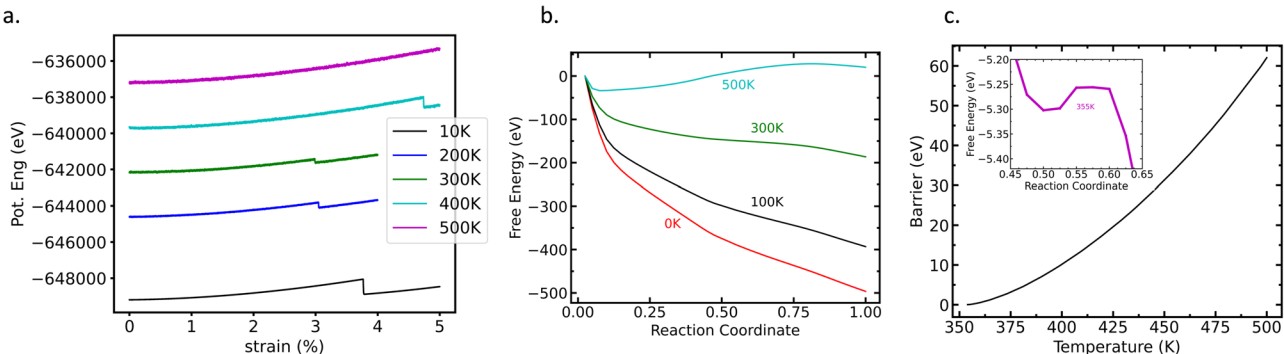

**Fig. 4 | Anti-Arrhenius kinetics under strain-controlled tensile loading.**
**a** Entropy assisted emergence of free energy barriers and corresponding nucleation kinetics lead to unconventional anti-Arrhenius behavior under a strain rate, i.e. beyond a certain temperature, the critical nucleation strain (c.f. jumps in the energy versus strain curves) increases. **b** For 5% tension i.e. beyond the athermal (3.8%) nucleation strain there is no energy barrier in the downhill steepest descent path as shown in red. As temperature increases the free energy landscape changes, and entropic barriers start to emerge. This is shown in (**c**). Inset one of the lowest temperature (355 K) path is shown where a barrier of 0.05 eV emerges. Such energetics is supported by our strain-rate MD simulation results (**a**) where at low temperature (200 K, blue) nucleation is energetic (lower than athermal strain) but at higher temperature (350 K onwards) there is no nucleation until 5% strain.

state at the minima and saddle points significantly differ. Situations involving dislocation reactions and nucleation events are natural candidates, given the long-range nature of the elastic interactions between dislocations and other micro-structural features. Strong non-Arrhenius effects can therefore be expected in many plastic deformation processes based on very general grounds. These have indeed been observed in computational studies of dislocation nucleation[11,14].

However, the difficulty of carrying out systematic campaigns with very well-characterized systems like defect-starved nanowires and micro-pillars is such that analogous experimental investigations are extremely rare. A notable exception is the work of Chen et al.[8], where the yield kinetics of nanowires loaded in tension were systematically explored at different temperatures and strain rates. This study highlights that the yield kinetics are consistent with extremely low prefactors $N_{sites}v_0$ on the order of $10^{-2}$/s, and ii) that tensile yielding can occur near, and even above, the athermal strength at finite temperature.

The first observation is directly consistent with the sharp decrease in vibrational entropy that is expected during dislocation nucleation, as discussed above. While the numerical values are not directly comparable, the observation of prefactors that are more than ten orders of magnitude smaller than standard values (even without accounting for the number of equivalent nucleation sites $N_{sites}$), is naturally explained by the vibrational stiffening that occurs for dislocation nucleation in materials loaded in tension. To address the second point, the rate theory model developed above was used to predict yield strains at different temperatures and strain rates, both in compression (c.f. Supplementary Fig. 2) and in tension (c.f. Supplementary Fig. 3). Analyzing the cumulative distribution of yield stress/strains (c.f. Supplementary Figs. 3 and 4) according to[5,8], the

variational model predicts very distinct temperature variations of the mean critical nucleation strain (Supplementary Figs. 3b and 4b) in different loading conditions. In compression, the mean compressive strain for nucleation steadily decreases with increasing temperature, as expected for conventional Arrhenius kinetics. In contrast, under tension, a non-monotonic trend is observed where the conventional decrease in nucleation strain with temperature is followed by an anti-Arrhenius increase of the yield strain with temperature (Supplementary Fig. 3b). A similar temperature dependence tensile was experimentally observed in the yield strengths of Cadmium, Zinc and Magnesium[29,30]. These observations are consistent with the results reported above (c.f., Fig. 4a), where the yield strain was observed to eventually exceed the athermal limit at high temperatures. Again, variational transition state theory provides a very natural explanation of this a priori non-intuitive effect.

A key feature of the entropic effects described in this work is the very strong asymmetry in the kinetic behaviors in tension and in compression, where entropic contributions either accelerate or suppress the nucleation kinetics, respectively. The second key feature is that the magnitude of these effects is strongly dependent on the strain imposed on the system, through the variations in the size of the critical loops: the lower the (absolute) strain, the larger the critical loops, the larger the entropic effects. Both these features can be observed in Table 1 which quantifies the variation in the harmonic TST parameters as a function of strain both in compression and in tension. For example, under compression, the apparent prefactors dramatically decrease from astronomical values, $10^{124}$ s$^{-1}$ at-2%, to merely very large values of $10^{21}$ s$^{-1}$ at-5%, which qualitatively corresponds to a compensation effect, where large barriers are partially offset by large prefactors. This enables nucleation events with very large static barriers (6.8 eV at-2%) to be observed in MD simulations. In contrast, in tension, prefactors are strongly suppressed, ranging from 2.6 s$^{-1}$ at 2% to $10^{10}$ s$^{-1}$ at 3%. In this case, prefactors decrease as barriers increase, a type of inverse compensation, leading to extremely low overall nucleation rates at small strains, which would be difficult to even observe experimentally even at high temperature, once the non-Arrhenius corrections are considered (c.f., Fig. 3).

Finally, it should be noted that the full variational TST treatment advocated here captures a much richer set of physics compared to alternatives, such as the Meyer-Neldel (MN) compensation rule, that were used in the past to analyze nucleation kinetics[31]. The MN rule[3,5,8,9,20] posits that the activation entropy ($\Delta S$) is related to the static energy barrier ($\Delta E$) through the simple relationship $\Delta S-\Delta E/T_{MN}$, where $T_{MN}$ is a positive constant. This empirical relation was recently

**Table 1 | Summary of the harmonic TST parameters for dislocation nucleation under tension and compression**

| Compression | | | Tension | | |
|---|---|---|---|---|---|
| Strain | $v_0$ (s$^{-1}$) | $\Delta E$ (eV) | Strain | $v_0$ (s$^{-1}$) | $\Delta E$ (eV) |
| −2% | $10^{124}$ | 6.8 | 2% | 2.59 | 0.73 |
| | | | 2.5% | $10^7$ | 0.37 |
| −3% | $10^{30}$ | 2.11 | 3% | $10^{10}$ | 0.15 |
| −5% | $10^{21}$ | 0.49 | | | |

As shown in the text, variational corrections to hTST must be considered at high temperatures, especially at low strains.

rationalized in terms of changes in vibrational frequencies between minima and saddle points[32], which share the same fundamental origin as the effects discussed here. However, MN theory is not sufficiently flexible to capture the full breadth of physics observed here, as it fails to predict (1) the temperature dependence of the (variational) activation entropy which leads to non-Arrhenius effects, (2) the reduction of the activation entropy with an increasing barrier, as observed in the tensile loading case (c.f., Supplementary Fig. 4).

A simple analytical alternative that captures these effects can be obtained by estimating the kinetic effect of the variational free energy barrier through a low-order expansion around the zero-temperature solution (c.f. Section 1.5). For a given pathway, it predicts a correction to the standard HTST of the form:

$$k_{\mathrm{vTST}} = k_{\mathrm{HTST}} e^{-\frac{T}{T_m}}, \tag{5}$$

where $T_m$ is a constant with units of temperature that can be related to the curvature of the potential energy at the saddle point and to the rate of change of the entropy along the minimum energy pathway. $T_m$ is therefore specific to each pathway and strain state and acts as a characteristic temperature scale above which variational TST predictions start to significantly differ from conventional HTST predictions. In the absence of a detailed thermodynamic analysis, $T_m$ can be considered as a free parameter that can be fitted to data.

Figures 2 c and 3b (dash-dotted lines) show that this simple analytical model quantitatively captures the non-Arrhenius corrections to the nucleation rate in a broad range of conditions, except at very low strains and high temperatures, where a higher order expansion of the free energy about the energy saddle becomes essential. This physics-based model can therefore be expected to provide a useful description of variational correction in a broad range of thermally activated dislocation reaction processes e.g., kink pair nucleation and migration in BCC, cross-slips, twin migration, junction formation, slip transmission across grain-boundaries, where non-Arrhenius corrections are expected.

### Effects of stress-controlled loading on kinetics

The results presented above were all obtained in a displacement-controlled setting when the volume of the simulation cell is held constant. It is often of interest to applications to also consider the constant stress setting. On general grounds, it could be expected that the magnitude of the entropic effects could be reduced[33], as the local strain states of the initial and final states would become largely equivalent, except for the effect of the stress concentration (now higher) at the step. Hence one could expect that the vibrational entropy in the initial and final states (after the emission and subsequent absorption of a full loop at the opposing surface) to be very similar, in contrast to the constant volume case where some amount of local strain would be released by the emission and absorption of a loop. Note however that the kinetics are controlled by the entropy change between the initial state and the dividing surface, hence a complete cancellation of the entropic effects is not expected.

To assess the difference between the two loading conditions, a perturbative approach (c.f. methods??) was employed to translate the constant-displacement results into a corresponding constant-stress scenario. To do so, it was assumed that the nature of the transition pathway itself would remain unchanged so that the internal coordinates of all atoms in each image of the NEB could be preserved. However, to account for the volume relaxation, the simulation cell was allowed to vary independently for each image so as to minimize its enthalpy at the stress corresponding to the initial state, which was carried out using a simplex approach.

The results, shown in Supplementary Figs. 6 and 7 for stresses corresponding to the 2% compression and 2% tension cases, respectively, demonstrate that the qualitative features of the thermodynamics and kinetics remain, but that the magnitude of the entropic acceleration (suppression) under compression (tension) decreases. For example, under compression, the harmonic prefactor reduces from $10^{124}$ s$^{-1}$ to $10^{39}$ s$^{-1}$ while the static, zero-temperature, barrier is mostly unchanged. Similarly, under the stress corresponding to 2% tension, the transition rates increase by 5 orders of magnitude, from $10^{0.4}$ to $10^{5.4}$ s$^{-1}$, again with minimal changes to the static barrier. Therefore, while the magnitude of the anomalous entropic effects decreases when constant stress is applied instead of a fixed volume distortion, they remain sufficiently strong to affect transition rates by many orders of magnitude.

We have shown that strong entropic effects arising from the anharmonic dependence of the vibrational frequency on the strain state of a material can have colossal effects on the nucleation kinetics in systems where the strain field in the material significantly varies between the minimum and transition state. Depending on the nature of strain changes, collective softening or stiffening of vibrational modes can lead to dramatically different behaviors, leading to extremely large or small prefactors, and to strongly non-Arrhenius kinetics that can only be captured by variational transition state theories. The analysis suggests that similar effects can be expected for a range of dislocation reactions so great care must be taken when interpreting the kinetics of such reactions.

## Methods

### Atomistic configuration

We consider dislocation nucleation at free surfaces in uniaxially-loaded fcc copper. To favor nucleation on a single slip system ($[1\bar{1}0](111)$) and reduce cross-slip and Escaig stress effects under axial loading conditions (the type of the loading is strain-controlled unless explicitly stated otherwise), we orient the primary slip-system at 45° (i.e. Schmid factor = 0.5) with respect to the horizontal loading axis ($x$), resulting in the y-axis of the supercell being aligned along the $[11\bar{2}]$ direction. For the large-scale MD simulations, we use a simulation cell of size $1452\text{Å} \times 1063\text{Å} \times 290\text{Å}$. For the minimum energy pathway calculations, we consider a smaller model with dimensions $172\text{Å} \times 177\text{Å} \times 70\text{Å}$. We verified that the energy path and barriers are quantitatively agnostic to box sizes in this range. A vacuum region of $50-100\text{Å}$ is introduced the simulation cells along the free surface-normal direction ($z$). To induce a heterogeneous nucleation site, a surface step of height $4b$ is introduced on the free surface, in a configuration that was previously used in atomistic studies[3]. Periodic boundary conditions are imposed in the $x$ and $y$ directions. We consider copper as modeled using the EAM potential due to Mishin et al.[28].

### Large-scale molecular dynamics simulations

A series of large-scale MD simulations on the large cells containing 38 million atoms described above was performed. In these simulations, the temperature is slowly increased at a rate of 10K/ns using a Langevin thermostat with a timestep of 1 fs, for a duration of 5 ns starting from an initial temperature of 600K. This protocol is repeated 50 times to estimate the nucleation time distribution. MD simulations under an imposed strain rate of $2 \times 10^8$/s were also carried out to quantify the anomalous kinetic behavior of dislocation nucleation under tensile loading for temperatures ranging between 10K and 500K. All MD simulations were executed using the open-source MD package LAMMPS[34].

### Minimum energy path computation

We perform CI-NEB[35] computations which are seeded with a chain of 30 replicas linearly interpolating between the initial pristine configuration and a half-loop configuration extracted from an MD trajectories after de-thermalizing by running 100 steps of minimisation via the FIRE algorithm[36]. To avoid extremely long transition paths that would lead to poor resolution around the saddle region, the atoms in

the final replica are held fixed, and a 2-step NEB relaxation process is employed. First, the force tolerance is set to $10^{-3}$ eV/Å with parallel and perpendicular spring constants of 1 eV/Å² and 5 eV/Å². This is followed by another round of relaxation with a stricter tolerance of $10^{-5}$ eV/Å with spring constants of 0.5 eV/Å² (parallel), and 1 eV/Å² (perpendicular). The climbing image is only activated in the second step.

To estimate energetics for constant stress ($\sigma^{prescribed}$) ensembles, it was assumed that the nature of the transition pathway itself would remain unchanged, so that the internal coordinates of all atoms in each image of the NEB could be preserved. However, to account for the volume relaxation, the simulation cell was allowed to vary independently for each replica so as to minimize $H(\sigma, \epsilon) = U - V_0 \sigma_{ij} \epsilon_{ij}$ at an applied external stress corresponding to the initial supercell ($V_0$). The minimization was per performed using an implementation[37] of Neldel-Mead simplex method. Efficacy of this approach is shown in Supplementary Fig. 6c and 7c, by monitoring the stress evolution along the nucleation pathway.

### Likelihood analysis of TST parameters

To analyze the nucleation event statistics of the large-scale MD simulations under an imposed temperature ramp, we derive the likelihood function of the parameters $\nu$ and $E_b$ of an effective TST model. In this setting, the instantaneous TST rate is given by $k(t, \nu, E_b) = N\nu e^{-\frac{E_b}{k_B(T_0 + T_r t)}}$, where $T_0$ is the initial temperature and $T_r$ is the temperature increment rate i.e., 10K/ns. This leads to the probability distribution for nucleation events ($t_{nuc}^i$) in $i$-th trajectory as

$$p^i(t_{nuc}^i, \nu, E_b) = \frac{e^{-\int_0^{t_{nuc}^i} k(s)ds}}{\int_0^{\infty} e^{-\int_0^t k(s)ds} dt} \qquad (6)$$

The joint likelihood of the effective parameters $\nu$ and $E_b$ given the specific set of nucleation times observed in the $M$ MD trajectories is then given by:

$$L(\nu, E_b | \{t_{nuc}\}) = \prod_{i=0}^{M} p^i(t_{nuc}^i, \nu, E_b). \qquad (7)$$

This likelihood is explicitly numerically maximized to identify the most statistically probable effective parameters (c.f., Fig. 1a).

### Analytical rate model for variational kinetics

To incorporate the effect of the variational temperature dependence of the free energy barrier, we develop a simple model where the potential energy and entropy along the reaction coordinate $\xi$ are expanded to the first non-trivial order around the energy saddle (which is the optimal dividing surface at zero temperature), i.e., $V(\xi) \simeq V_{\xi_{saddle}} + \frac{1}{2}\frac{\partial^2 V}{\partial \xi^2}(\xi - \xi_{saddle})^2$ and $S(\xi) = S_{\xi_{saddle}} + \frac{\partial S}{\partial \xi}(\xi - \xi_{saddle})$.

This expansion admits a closed-form solution for the temperature-dependent variational free-energy barrier of the form

$$\Delta F_{vTST}(T) = \Delta F_{HTST} - \frac{S_\xi^2}{2 V_{\xi\xi}} T^2, \qquad (8)$$

where $\Delta F_{HTST} = V_{saddle} - TS_{saddle}$ is the static, 0K, HTST barrier, $S_\xi = \frac{\partial S}{\partial \xi}$, and $V_{\xi\xi} = \frac{\partial^2 V}{\partial \xi^2}$.

Defining an effective temperature $T_m = -\frac{2k_B V_{\xi\xi}}{S_\xi^2}$ and injecting the variational barrier $\Delta F(T)$ into a TST rate expression, the nucleation rate can now be simply expressed as

$$\begin{aligned} k_{vTST} &= N_{sites} \nu_0 e^{-\frac{F_{vTST}}{k_B T}} \\ &= N_{sites} \nu_0 e^{-\frac{F_{HTST}}{k_B T}} e^{-\frac{T}{T_m}} \\ &= k_{HTST} e^{-\frac{T}{T_m}}, \end{aligned} \qquad (9)$$

i.e., the first-order variational correction to the free-energy barrier at finite temperature is simply given by the HTST rate multiplied by a correction that exponentially suppresses the rate as a function of $T/T_m$. This reflects the fact that a first-order variational correction to TST can only *increase* the free-energy barrier, and hence *decrease* the rate compared to the baseline HTST result, which is itself recovered in the low-temperature limit. Qualitatively, $T_m$ sets a temperature scale below which HTST is expected to be accurate, but above which variational corrections significantly lower the HTST prediction. As shown in Figs 2c and 3b, the model predictions closely follow the ones obtained from an explicit variational TST treatment, except at low strain and high temperatures where higher order corrections are required to capture the pronounced temperature dependence of the location of the variational dividing surface. Note that this expression applies for a given energy/entropy landscape (i.e., at fixed applied strain); to capture the strain dependence of the correction w.r.t. HTST one needs to account for the variation of $T_m$ with strain. As shown in Supplementary Fig. 5, $T_m$ was empirically observed to vary linearly over the range of strain that was considered. However, within the low-order expansion used here, $T_m$ should remain non-negative, and so the regime of validity of a linear approximation of $T_m$ vs strain is necessarily limited and should be validated with care.

## Data availability

Source data are provided with this paper. Representative simulation data and analysis examples can be found at https://github.com/BagchiS6/Variational_Strain. codes ca/codes can only be provided upon reasonable request to the corresponding author. Source data are provided with this paper.

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

## Acknowledgements

We thank Enrique Martinez for the fruitful discussions. We gratefully acknowledge the support provided by the Laboratory Directed Research and Development program of Los Alamos National Laboratory (LANL) under Projects LDRD 2020057DR (initial stages of development) and LDRD2022063DR (variational analysis). This work used computing resources provided by the LANL Institutional Computing Program and NESAP program of NERSC. LANL is operated by Triad National Security, LLC, for the National Nuclear Security Administration of U.S. Department of Energy (Contract No. 89233218CNA000001).

## Author contributions

S.B. and D.P. designed the research. S.B. in consultation with D.P. derived the numerical and analytical results, ran the atomistic simulations and performed the analysis. S.B. and D.P. wrote the paper.

## Competing interests

The authors declare no competing interests.
