## [Transparent Peer Review File · Nature Communications]

Anomalous entropy-driven kinetics of dislocation nucleation

Corresponding Author: Dr Soumendu Bagchi

Version 0:

Reviewer comments:

Reviewer #1

(Remarks to the Author)

I really enjoyed reading this since it presents a clean example of dislocation nucleation behaving in a way that could then be summarized by a new transition state theory approach.

The only thing I would suggest would be quantitative comparisons of the nucleation rate compression vs tensile case.

Reviewer #2

(Remarks to the Author)

The study meticulously investigated the origin of the entropic effect on the rate of dislocation nucleation using large-scale MD simulations and variational reaction theory, and explored its asymmetric nature under conditions of compression and tension. This research provides clear insights into the entropic effect within thermally-activated, dislocation-related processes, representing a significant contribution to this field of study. I highly recommend the publication of this article, contingent upon addressing the following issues to enhance the clarity of the presentation.

- 1) In the derivation of analytical expressions as well as in large-scale MD simulations, the authors should more clearly specify and clarify the loading conditions. The estimated entropic effect can vary significantly depending on whether the conditions are (i) displacement-controlled or (ii) stress-controlled. (Ref: <https://doi.org/10.1073/pnas.1017171108>, <https://doi.org/10.1557/jmr.2011.275>)
- 2) Building on the first point, conducting both stress-controlled (constant stress) and strain-controlled (constant strain) MD simulations to estimate the nucleation rate at constant loading condition would provide valuable insights into the previously mentioned aspects.
- 3) Regarding the CI-NEB method, it appears that simulations are conducted under constant strain conditions. It would be important to examine how much stress changes during the formation of a dislocation loop. Using a significantly longer simulation cell (along the loading direction) could reduce the amount of compressive or tensile strain relaxation during the dislocation nucleation loop formation, potentially diminishing the observed asymmetry between compression and tension in the very long nanowire limit. I recommend conducting additional NEB simulations and per-atom entropy contribution calculations using nanowires that are twice and four times as long, to explore how this size variation influences the compression-tension asymmetry in entropic effect.
- 4) In the study of nucleation during first-order phase transitions, it has been reported that the entropic effect on the free energy barrier includes a corrective logarithmic ($\log(n)$) term. (Ref: <https://doi.org/10.1103/PhysRevLett.21.973>, <https://doi.org/10.1103/PhysRevE.81.030601>) Since the dislocation nucleation process involves the formation of critical-sized loops, and shares similarities with two-dimensional nucleation phenomena observed in the 2D Ising model, a comparison between the free energy calculated in this study and Langer's free energy barrier expression, which includes the $\log(n)$ term, would be intriguing.

Version 1:

Reviewer comments:

Reviewer #1

(Remarks to the Author)

I am happy with the comprehensive response of the authors and thus recommend publication.

Reviewer #2

(Remarks to the Author)

The authors have addressed my concerns with exceptional detail. The revised manuscript is now considerably more comprehensive, and I am pleased to recommend it for publication.

Response to the reviewers' comments

Reviewer #1 (Remarks to the Author):

I really enjoyed reading this since it presents a clean example of dislocation nucleation behaving in a way that could then be summarized by a new transition state theory approach.

We thank the reviewer for their positive feedback. Likewise, we strongly think that such an approach involving variational rate theory along with an accurate estimation of entropic effects and a clean modeling example depicting the anharmonic effects of dislocation nucleation will open the doors for several detailed investigations concerning thermal activation of reactions crucial in crystal plasticity (e.g., dislocation glide [Allera, Swinburne et al., 2024], cross-slip [Yang, Cai 2023], junction formation as well interactions with grain boundaries).

The only thing I would suggest would be quantitative comparisons of the nucleation rate compression vs tensile case

This is an important point, as a key feature of our analysis is a strong asymmetry between the behavior in tension and compression, which could potentially be observed experimentally, although, as we point out, systematic experimental studies are scarce. To address this point, we added the following paragraph to the Discussion section to describe the tension/compression asymmetry more systematically.

Compression			Tension		
Strain	v_0 (s^{-1})	ΔE (eV)	Strain	v_0 (s^{-1})	ΔE (eV)
-2%	10^{124}	6.8	2%	2.59	0.73
			2.5%	10^7	0.37
-3%	10^{30}	2.11	3%	10^{10}	0.15
-4%	10^{22}	0.58			
-5%	10^{21}	0.49			

Table 1: Summary of the harmonic TST parameters for dislocation nucleation under tension and compression. As shown in the text, variational corrections to hTST must be considered at high temperatures, especially at low strains.

A key feature of the entropic effects described in this work is the very strong asymmetry in the kinetic behaviors in tension and in compression, where entropic contributions either accelerate or suppress the nucleation kinetics, respectively. The second key feature is that the magnitude of these effects is strongly dependent on the strain imposed on the system, through the variations in size of the critical loops: the lower the (absolute) strain, the larger the critical loops, the larger the entropic effects. Both these features can be observed in Table 1 which quantifies the variation in the harmonic TST parameters as a function of strain both in compression and in tension. For example, under compression, the apparent prefactors dramatically decrease from astronomical values, $10^{124} s^{-1}$ at -2%, to merely very large values of $10^{21} s^{-1}$ at -5%, which qualitatively corresponds to a compensation effect, where large barriers are partially offset by large prefactors. This enables nucleation events with very large static barriers (6.8 eV at -2%) to be observed in MD simulations. In contrast, in tension, prefactors are strongly suppressed, ranging from $2.6 s^{-1}$ at 2% to $10^{10} s^{-1}$ at 3%. In this case, prefactors decrease as barriers increase, a type of inverse

compensation, leading to extremely low overall nucleation rates at small strains, which would be difficult to even observe experimentally even at high temperature, once the non-Arrhenius corrections are considered (c.f., Fig.3).

Reviewer #2 (Remarks to the Author):

The study meticulously investigated the origin of the entropic effect on the rate of dislocation nucleation using large-scale MD simulations and variational reaction theory, and explored its asymmetric nature under conditions of compression and tension. This research provides clear insights into the entropic effect within thermally-activated, dislocation-related processes, representing a significant contribution to this field of study. I highly recommend the publication of this article, contingent upon addressing the following issues to enhance the clarity of the presentation.

We are grateful to the referee for their careful reading and constructive suggestions. We provide a point-by-point response below.

1) In the derivation of analytical expressions as well as in large-scale MD simulations, the authors should more clearly specify and clarify the loading conditions. The estimated entropic effect can vary significantly depending on whether the conditions are (i) displacement-controlled or (ii) stress-controlled.

(Ref: <https://doi.org/10.1073/pnas.1017171108>, <https://doi.org/10.1557/jmr.2011.275>)

This is an important point, which we discuss extensively in the following. The first version of the manuscript focused exclusively on the displacement-controlled setting. However, the approach can easily be generalized to the stress-controlled setting. This has now been made clearer throughout the text.

2) Building on the first point, conducting both stress-controlled (constant stress) and strain-controlled (constant strain) MD simulations to estimate the nucleation rate at constant loading condition would provide valuable insights into the previously mentioned aspects.

3) Regarding the CI-NEB method, it appears that simulations are conducted under constant strain conditions. It would be important to examine how much stress changes during the formation of a dislocation loop.....(more follows later)

We thank the reviewer for pointing us to these thorough analyses on anharmonicity in dislocation reactions (<https://doi.org/10.1557/jmr.2011.275>) [1] under stress and strain-controlled loadings. We also note that we had previously referred to parts of these analyses and results as presented in: <https://doi.org/10.1073/pnas.1017171108> [2] in our submitted version of the manuscript.

In contrast to a constant strain setting, the extent of local volume relaxation, and hence of entropic contributions, due to the nucleation and growth of the loops can, under a constant stress ensemble, be expected to be reduced due to the volume response of the cell. This volume relaxation will render the local strain environments before and after the emission of a full loop mostly equivalent (modulo the effect of the change in stress concentration at the step), which could affect the magnitude of the entropic effects at the dividing surface, and hence the nucleation rates.

To capture the consequences of this difference on the kinetics, we rely on a perturbative approach around the constant-volume NEB analysis presented in the original manuscript. In this case, we assume that the transition pathway itself is largely similar in both ensembles so that the internal coordinates of the atoms in each of the NEB images can be held constant, while the cell degrees of freedom are optimized to minimize the Gibbs free energy along the path. This approach is now summarized in the updated Methods section. As shown in Fig. R1c, this leads to near constant stresses along the path.

The results for the stress state corresponding to the $\varepsilon_{xx} = 2\%$ compression strain are presented in Fig. R1 and in the updated manuscript. The Gibbs free energy, enthalpy (red) and entropy (blue) variation along the MEP are shown in Fig. R1a. The increase of entropy along the MEP is still observed at constant stress, although its magnitude is lower than at constant strain. The smaller entropic contribution is reflected in a comparatively smaller acceleration of the nucleation rates, as shown in Fig. R1b. Note that the entropic effects are still extremely significant at constant stress, with a predicted HTST prefactor of 10^{39} /s.

Figure R1 Stress controlled analysis of (Gibbs) free energy (a), rates (b), and change of stress state along MEP (c): Compression

Figure R2 Stress controlled analysis of (Gibbs) free energy (a), rates (b), and change of stress state along MEP (c): Tension

The equivalent analysis performed under tension is presented in Fig. R2. Again, the qualitative features of the free energy landscape and the effect on the kinetics remain similar to the constant-strain case, while the magnitude of the entropic kinetic suppression decreases by about 5 orders of magnitude compared to the constant displacement case.

This shows that while the thermodynamic ensemble does affect the magnitude of the entropic effects, the broad characteristics of the observed anomalous kinetics are expected to be robust across both strain-controlled and stress-controlled conditions.

A summary of the above discussion was added in the main text under an additional section “effect of stress-controlled loading” along with some of the figures presented here added to the Supplementary Materials:

The results presented above were all obtained in a displacement-controlled setting when the volume of the simulation cell is held constant. It is often of interest to applications to also consider the constant stress setting. On general grounds, it could be expected that the magnitude of the entropic effects could be reduced [34], as the local strain states of the initial and final states would become largely equivalent, except for the effect of the stress concentration (now higher) at the step. Hence one could expect that the vibrational entropy in the initial and final states (after the emission and subsequent absorption of a full loop at the opposing surface) to be very similar, in contrast to the constant volume case where some amount of local strain would be released by the emission and absorption of a loop. Note however that the kinetics are controlled by the entropy change

between the initial state and the dividing surface, hence a complete cancellation of the entropic effects is not expected.

To assess the difference between the two loading conditions, a perturbative approach was employed to translate the constant-displacement results into a corresponding constant-stress scenario. To do so, it was assumed that the nature of the transition pathway itself would remain unchanged, so that the internal coordinates of all atoms in each image of the NEB could be preserved. However, to account for the volume relaxation, the simulation cell was allowed to vary independently for each image so as to minimize its enthalpy at the stress corresponding to the initial state, which was carried out using a simplex approach.

The results, shown in SI Figures 6 and 7 for stresses corresponding to the 2% compression and 2% tension cases, respectively, demonstrate that the qualitative features of the thermodynamics and kinetics remain, but that the magnitude of the entropic acceleration (suppression) under compression (tension) decreases. For example, under compression, the harmonic prefactor reduces from 10^{124} s^{-1} to 10^{39} s^{-1} while the static, zero-temperature, barrier is mostly unchanged. Similarly, under the stress corresponding to 2% tension, the transition rates increase by 5 orders of magnitude, from $10^{0.4}$ to $10^{5.4} \text{ s}^{-1}$, again with minimal changes to the static barrier. Therefore, while the magnitude of the anomalous entropic effects decreases when a constant stress is applied instead of a fixed volume distortion, they remain sufficiently strong to affect transition rates by many orders of magnitude.

Using a significantly longer simulation cell (along the loading direction) could reduce the amount of compressive or tensile strain relaxation during the dislocation nucleation loop formation, potentially diminishing the observed asymmetry between compression and tension in the very long nanowire limit. I recommend conducting additional NEB simulations and per-atom entropy contribution calculations using nanowires that are twice and four times as long, to explore how this size variation influences the compression-tension asymmetry in entropic effect.

Model size effect on nucleation kinetics: To quantitatively probe the general effect of size variation of our atomistic models, we have performed multiple CI-NEB calculations to estimate the MEP and the activation parameters. We proportionally scale the volume not only in the loading direction (x) but also in the lateral direction (y) to account for the Poisson effects (we note that our systems are slabs of finite thickness with PBCs in x and y and open surfaces in z). As shown in Figure R3, nucleation kinetics remain largely unaffected by model size variation, providing further confidence that the results presented in the paper are robust. We also performed MD simulations at constant strain rate on different cell sizes and observed no systematic trend (c.f., Fig. R3a).

Figure R3 Model Size dependence on nucleation kinetics under compression: Simulations performed for different model sizes (4x4, 6x6, 8x8) upto 38 million atoms (200x200, with reference to our smallest model 1x1~ 0.18 million).

Likewise, under 2% tension, the anti-Arrhenius nucleation behavior reported in Section D. of the main text, remains consistent across model sizes ranging from 0.18 million (Fig. R4a) to 38 million (R4b). Low temperature nucleation at ~3% strain is suppressed at higher temperatures (e.g. 500K in R4a and R4b) in both the cases shown. This results from a negative entropy change which leads to an increase in or even emergence of free energy barriers.

Figure R4 Model size dependence on the anti-Arrhenius nucleation kinetics under tensile constant strain-rate simulations.

4) In the study of nucleation during first-order phase transitions, it has been reported that the entropic effect on the free energy barrier includes a corrective logarithmic ($\log(n)$) term. (Ref: <https://doi.org/10.1103/PhysRevLett.21.973>, <https://doi.org/10.1103/PhysRevE.81.030601>) Since the dislocation nucleation process involves the formation of critical-sized loops, and shares similarities with two-dimensional nucleation phenomena observed in the 2D Ising model, a

comparison between the free energy calculated in this study and Langer's free energy barrier expression, which includes the $\log(n)$ term, would be intriguing.

We thank the reviewer for pointing us to these interesting studies. In the current context, it appears difficult to unambiguously disentangle the origin of different contributions to the free energy. E.g., the elastic contribution alone are expected to exhibit a very complex dependence on imposed stress and on materials properties (<https://journals.aps.org/prb/pdf/10.1103/PhysRevB.78.064109>) even within linear elasticity. Also, in our analysis, only the vibrational contributions to the entropy are taken into account, not including possible shape fluctuation of the critical loop itself, which are the origin of the logarithmic corrections proposed by Langer. These fluctuations appear to be much less significant for dislocation half-loops that visually look very “smooth” even at finite temperature compared to what is observed in the 2D Ising model (c.f., Fig. 4b in <https://doi.org/10.1103/PhysRevE.81.030601>) where the critical nuclei depart significantly from the idealized circular shape assumed by the CNT. While this area of investigation is extremely interesting, the identification of possible anomalies in the free energy and their attribution to specific sources would require extensive analysis that are outside of the scope of this paper.